# The Interplay between cGMP and Calcium Signaling in Alzheimer’s Disease

**DOI:** 10.3390/ijms23137048

**Published:** 2022-06-24

**Authors:** Aileen Jehle, Olga Garaschuk

**Affiliations:** Department of Neurophysiology, Institute of Physiology, Eberhard Karls University of Tübingen, 72074 Tübingen, Germany; aileen.jehle@uni-tuebingen.de

**Keywords:** cyclic guanosine-3′,5′-monophosphate, nitric oxide, neurodegeneration, neuroinflammation, astrocytes, microglia, phosphodiesterase inhibitors

## Abstract

Cyclic guanosine monophosphate (cGMP) is a ubiquitous second messenger and a key molecule in many important signaling cascades in the body and brain, including phototransduction, olfaction, vasodilation, and functional hyperemia. Additionally, cGMP is involved in long-term potentiation (LTP), a cellular correlate of learning and memory, and recent studies have identified the cGMP-increasing drug Sildenafil as a potential risk modifier in Alzheimer’s disease (AD). AD development is accompanied by a net increase in the expression of nitric oxide (NO) synthases but a decreased activity of soluble guanylate cyclases, so the exact sign and extent of AD-mediated imbalance remain unclear. Moreover, human patients and mouse models of the disease present with entangled deregulation of both cGMP and Ca^2+^ signaling, e.g., causing changes in cGMP-mediated Ca^2+^ release from the intracellular stores as well as Ca^2+^-mediated cGMP production. Still, the mechanisms governing such interplay are poorly understood. Here, we review the recent data on mechanisms underlying the brain cGMP signaling and its interconnection with Ca^2+^ signaling. We also discuss the recent evidence stressing the importance of such interplay for normal brain function as well as in Alzheimer’s disease.

## 1. Introduction

Cyclic guanosine monophosphate (cGMP) is one of the ubiquitous second messengers of the body and brain. It targets a manifold of downstream pathways, thus eliciting a broad variety of cellular/tissue effects. The role of cGMP has been extensively studied in the periphery, where it has important physiological functions in the vasculature, heart, pulmonary arteries, and gastrointestinal tract [1]. The dysfunction of the aforementioned pathways (e.g., in the cardiovascular domain) is common and the development of drugs, targeting these pathways, has substantial therapeutic implications. The most known are phosphodiesterase (PDE) inhibitors (e.g., PDE5 inhibitor Sildenafil), used in erectile dysfunction [2]. In the brain, cGMP is involved in signal transduction in the retina as well as synaptic plasticity, learning and memory formation, activity-dependent hyperemia, and inflammation [3,4]. Therefore, it has gained attention as a possible druggable target for treating neurodegenerative diseases.

In neuro-glial brain networks, the production of cGMP is modulated in an activity-dependent manner, with increases in synaptic transmission as well as second messenger-mediated rises in the cytosolic free Ca^2+^ concentration ([Ca^2+^]_i_) enhancing the production of cGMP [5,6,7]. Both neuronal activity and Ca^2+^ signaling are affected in Alzheimer’s disease (AD) patients or mouse models of the disease [7,8,9]. In fact, AD patients are known to have an increased incidence of epileptic seizures, which is independent of the disease stage and highest in cases with early disease onset [7,10,11,12]. The vicinity of amyloid plaque deposits, studied in animal models of the disease, is known for its profound neuronal, astrocytic, and microglial hyperactivity [8,10,12]. AD is the most common form of dementia, and it is expected that the number of AD cases will rise significantly in the years to come due to the aging of the Western population, consequences of the COVID-19 pandemic, and profound inflammation/brain traumata obtained during war [13,14,15], making this disease a major concern for modern society.

The cGMP signaling pathway is reportedly affected during the early stages of AD [16], making it an interesting target for developing new treatment strategies. In this review, we summarize the physiological roles of the cGMP signaling pathway in different constituents of the neuro-glial brain network, its interconnection with Ca^2+^ signaling, its implications in AD, and the current attempts in developing new AD therapies targeting the cGMP signaling pathway.

## 2. cGMP Toolkit in the Brain

There are two main pathways for cGMP production (Figure 1). The first pathway is mediated by NO. This short-lived free radical molecule is mainly produced through nitric oxide synthases (NOS). Like other tissues, the brain contains two types of the constitutively expressed NOS (neuronal (nNOS) and endothelial (eNOS)) as well as the inducible NOS (iNOS), whose expression is triggered by different stimuli. The nNOS is widely expressed throughout the brain and is abundant in neurons, but it is also found in other brain cells such as astrocytes and endothelial cells. eNOS is most commonly expressed in endothelial cells [17], but also in blood vessel-associated astrocytes as well as in the layer II–IV cortical neurons [18,19]. The activity of both types of constitutively expressed NOS is regulated by Ca^2+^ and calmodulin [20,21]. This mechanism of activation enables a fast and transient increase in the intracellular NO concentration in response to cells’ activity. In contrast to constitutively expressed NOS isoforms, the expression of iNOS is triggered. This form of NOS is part of the immune response machinery and therefore is mainly found in astrocytes and microglia [22]. To a smaller degree, it is also expressed in neurons and endothelial cells of the blood–brain barrier (BBB) [23]. The main stimuli inducing iNOS activity are tumor necrosis factor-alpha (TNF-α), interleukin-1β (IL-1β), interferon-gamma (IFN-γ), lipopolysaccharide (LPS), and double-strand RNA (dsRNA) [22,24,25]. The subsequent NO production is primarily regulated at the transcriptional level through binding sites for the nuclear factor κB (NF-κB) and the phosphorylated cyclic adenosin monophosphate (cAMP) response element-binding protein (pCREB) in the promoter region of the iNOS gene [26,27]. Once the expression of iNOS is triggered, it generates NO over a prolonged period of time. One factor controlling the expression of iNOS via NF-κB is the nuclear factor erythroid 2-related factor 2 (Nrf2). This transcription factor can be activated by NO [28,29] and causes, among others, the expression of heme oxygenase 1. This protein is known to inhibit NF-κB, thus causing a NO-coupled reduction in iNOS expression in case of inflammation (Figure 1).

As a gaseous molecule, NO can induce both an intracellular as well as an intercellular response by triggering the production of cGMP by the soluble guanylate cyclases (NO-GCs). Depending on the cell type, the NO-GC is predominantly cytosolic, or predominantly membrane-associated, as, for example, in neurons [30]. The enzyme contains two subunits, a longer β-subunit and a shorter α-subunit, together forming an α/β-heterodimer, which is required for catalytic activity [31]. There are two subtypes of NO-GC. Both subtypes contain four structures: a coiled-coil domain, a Per-Arnt-Sim (PAS) domain, the heme-nitric-oxide oxygen binding (H-NOX) domain, and the catalytic domain (CAT) [32,33,34,35]. NO binds to the H-NOX domain, leading to a conformational change, allowing the catalytic domain to produce cGMP from GTP [36]. NO-GC-1, formed by an α1β1-heterodimer, is widely expressed throughout the organism, whereas the NO-GC-2 (α2β1-heterodimer) is strongly expressed in the CNS [37].

The second pathway of cGMP production is triggered by the activation of natriuretic peptide receptors (NPRs). Three types of these receptors, all having an extracellular binding domain for natriuretic peptides (NPs), are known to date. Two subtypes have an intracellular guanylate cyclase domain [38]. These are the NPR-A (also called particulate NPR1 or pGC-A), activated by atrial natriuretic peptide (ANP), and brain-type natriuretic peptide (BNP) as well as the NRP-B (also called NPR2 or pGC-B) activated by C-type natriuretic peptide (CNP). NPR-C (also called NPR-3) has no intracellular catalytic domain and no enzymatic activity. Its main function is the clearance of NP from the extracellular fluid [39]. There is also evidence that NPR-C can activate G-proteins, which inhibit adenylyl cyclase activity.

NPs are expressed in many tissues including the brain, with different distributions in different brain areas. NPR-B is ubiquitously expressed in neurons and glial cells throughout all brain areas (gene name *npr2,* available from www.proteinatlas.org, accessed on 23 March 2022) [40]. NPR-A has very low expression within the brain, but it can be found within endothelial cells (gene name *npr1*, available from www.proteinatlas.org, accessed on 23 March 2022) [40]. ANP and BNP are mostly expressed in muscle tissue, but in low concentrations; they can also be found in the CNS [41,42,43,44]. Although proteomic profiling shows very low NPR-A in the brain, there is some evidence for ANP signaling via NPR-A in astrocytes [45]. The third natriuretic peptide CNP has the strongest expression in the brain (10 times higher than that for ANP or BNP [46]). The expression is widespread throughout the different brain regions and is mainly localized to inhibitory interneurons and glial cells (gene name *nppc*, available from www.proteinatlas.org, accessed on 23 March 2022) [40], suggesting that CNP signaling is the major NP signaling type in the brain.

cGMP can modulate the activity of three different proteins (Figure 1). One of these downstream targets is the cGK (protein kinase G), a member of the family of serine/threonine protein kinases. Generally, there are two different cGKs, encoded by the *prkg1* and the *prkg2* genes. cGK-I has two splice variants, cGK-Iα and cGK-Iβ. All cGK isoforms are widely distributed throughout the brain, with different distributions in different brain areas [47,48,49]. Upon cGMP binding, the phosphotransferase activity of cGK is increased, leading to the phosphorylation of downstream targets [50,51]. cGK has a regulatory domain that can be autophosphorylated [52], leading to activation of the enzyme [53]. Phosphorylation by cGK has been implicated in many brain functions, including anxiety and fear behavior [54,55], generation of the circadian rhythm [55,56], and long-term potentiation [57,58].

The cyclic nucleotide-gated ion channels (CNGCs) are the second target of cGMP. Although part of the voltage-gated ion channel superfamily, these channels possess functional properties of ligand-gated ion channels. They have an intracellular ligand-binding domain, which can be accessed by both cGMP and cAMP, leading to an influx of Na^+^ and Ca^2+^ and an efflux of K^+^ [59]. CNGCs were first found in cone photoreceptors [60], but further studies have identified CNCGs in many different tissues, including the brain, where the CNGC activation by cGMP modulates glutamatergic signaling [61].

The third class of proteins affected by cGMP are phosphodiesterases (PDEs; Figure 1). These enzymes are responsible for the degradation of cyclic nucleotides. To date, 11 types of PDEs have been identified. Some PDEs are specific for cAMP (PDE4, 7, and 8) and others for cGMP (PDE5, 6, and 9). The rest of the PDEs are dual-substrate PDEs, able to hydrolyze both cAMP and cGMP. Most PDEs can be found in the CNS, with different expression levels in different brain regions [62,63]. The cGMP-specific PDE6 is only expressed in the retina, whereas PDE5 and PDE9 are commonly found in the CNS. cGMP itself can control some of the PDE activity (Figure 1). For example, a low concentration of cGMP is able to decrease PDE3 activity, allowing for an increase in cAMP concentration, whereas a higher cGMP concentration leads to an increase in PDE2 activity, reducing cAMP concentration, and to a smaller extent, also the cGMP concentration [64]. PDE5 activity is also regulated by cGMP through cGK-I phosphorylation [65]. Interestingly, the cGMP-specific PDE5 and PDE9 seem to control different subcellular pools of cGMP. PDE5 mainly controls the cGMP pool formed through NO/NO-GC activation, whereas PDE9 controls the NP/NPR pool of cGMP [66].

The NO/cGMP/cGK pathway has autoregulatory capabilities. As a phosphokinase, cGK can regulate the activity of other enzymes. Some of the targets for cGK belong to the NO/cGMP/cGK pathway. For example, cGK can decrease the activity of NO-GC, serving as a negative feedback mechanism for cGMP production [67,68]. cGK also changes PDE5 activity (Figure 1). By phosphorylating this enzyme, the affinity for cGMP increases, leading to faster degradation of cGMP and therefore a faster termination of the cGMP-mediated signaling [67].

Taken together, brain cells possess a complex and versatile cGMP toolkit, capable of monitoring changes in neural network activity as well as a release of multiple modulatory substances (Figure 1). Via numerous, and in many instances, cell type-specific (see below), interactions between different components within and outside of the NO/cGMP/cGK pathway, the cGMP signaling is likely to play an important, although not yet completely understood, role in brain function.

## 3. Cell Type Specificity of the cGMP Pathway

The NO/cGMP/cGK pathway has been implicated in many processes within the brain, such as modulation of synaptic transmission, neuroprotection, and immune response. In these processes, cGMP can have a beneficial or a detrimental effect, depending on its concentration and the cell type under study. Since the possible effects of cGMP are diverse, it is worth taking a closer look at cell type-specific cGMP signaling pathways in the most important cell types of the brain (Figure 2).

### 3.1. cGMP Signaling in Neurons

NOS, NO-GC, and cGK can be found in neuronal somata, dendrites, and in dendritic spines, pointing toward the important role of the NO/cGMP/cGK pathway in synaptic transmission. A well-known role of cGMP is its involvement in the long-term potentiation of synaptic transmission, a key process for learning and memory formation. An increase in cGMP production was observed in response to the activation of glutamatergic *N*-methyl-*D*-aspartate (NMDA), cholinergic and dopaminergic receptors [69,70,71,72]. At glutamatergic synapses, both nNOS and NO-GC-2 were shown to bind to the PDZ domain of post synaptic density protein 95 (PSD-95), a major scaffolding molecule enriched at glutamatergic synapses [30,73]. This proximity allows for the fast and specific relay of the information from the level of pre-/postsynaptic activity to the NO/cGMP/cGK pathway. Strong neuronal activation and hereby induced NMDA receptor-mediated Ca^2+^-influx lead to the activation of nNOS [5,6], an increase in the intracellular NO concentration, activation of NO-GC, production of cGMP, and activation of cGK, which can phosphorylate downstream targets. In conditions of the LTP induction, there are two main functions of cGK (Figure 2). Activated cGK-I can translocate to the nucleus, where it phosphorylates CREB, thus inducing the transcription of LTP-relevant genes [74,75,76]. Furthermore, cGK-II can phosphorylate GluA1, an α-amino-3-hydroxy-5-methyl-4-isoxazolepropionic acid (AMPA) receptor subunit, leading to more AMPA receptors being included in the postsynaptic density [77,78].

Next to the postsynaptic function, the pathway is also involved in the presynaptic part of LTP (Figure 2). nNOS and NO-GC can be found in both pre- and postsynaptic structures. NO-GC is mostly found presynaptically, whereas nNOS is mainly found in the postsynaptic density [79]. Interestingly, the presynaptic structures containing NO-GC often oppose nNOS-containing postsynaptic structures [79]. The role of NO as a retrograde messenger between pre- and postsynaptic structures has been well established [80,81,82]. It is either produced by NMDA receptor-coupled nNOS, or by eNOS [83,84], which can be, for example, found in postsynaptic structures of cortical neurons [85]. Two NO-GC isoforms are expressed in the CNS. NO-GC-1 is mostly found in presynaptic structures, whereas NO-GC-2 is more commonly found postsynaptically [79,83]. Consistently, the NO-GC-1 stimulation was shown to be involved in the NO/cGMP-mediated increase in the presynaptic glutamate release in hippocampal CA1 neurons [83]. Increased vesicle release is likely induced by activation of hyperpolarization-activated CNGCs (Figure 2) [83]. During prolonged synaptic stimulation, the activation of cGK is needed for vesicle recycling, most likely by increasing the amount of phosphatidylinositol-4,5-bisphosphate (PIP_2_, Figure 2) [86,87,88].

cGMP is also involved in structural plasticity through the activation of cGK, and the cGK-mediated phosphorylation of downstream effectors, such as vasodilator-stimulated phosphoprotein (VASP) and the Ras homologue-GTPase A (RhoA) [89]. Depending on their phosphorylation status, both proteins can alter actin remodeling. VASP is an actin-binding protein, inducing actin assembly and thus influencing the density, size, and morphology of synapses [90]. It was shown that spine development is positively influenced by VASP through NO/NO-GC signaling [91]. RhoA is another protein influencing actin remodeling. RhoA signaling is associated with dendrite retraction and inhibition of synapse formation [92]. Activation of the NO/cGMP/cGK signaling pathway can induce RhoA-mediated loss of synapses [93]. Therefore, cGMP signaling can have both a beneficial and a detrimental effect on spine stability.

Taken together, cGMP signaling is involved in several aspects of LTP. It can influence early LTP through presynaptic modulation of vesicle release and affect late-phase LTP through the induction of gene expression and synaptic stability. These processes are all tightly regulated, and their imbalance can have a detrimental effect on synaptic signaling.

### 3.2. cGMP Signaling in Astrocytes

Astrocytes play a key role in brain homeostasis, including buffering of extracellular K^+^ and pH, regulation of synaptic transmission as well as water transport, maintenance of the blood-brain barrier, and providing energy substrates to neurons. Additionally, they are part of the immune defense of the brain.

Astrocytes express many elements of the cGMP signaling pathway, including all three NOS isoforms (Figure 2). However, nNOS is found in much lower concentrations in astrocytes than in neurons and at a different subcellular location. In neurons, nNOS is coupled to NMDA receptors via the PDZ-binding domain (see above), whereas in astrocytes, nNOS is mostly found in the cytosol [94]. eNOS, the second type of Ca^2+^-activated NOS, is mainly found in blood vessel-associated astrocytes [18,19]. The expression of iNOS, the inducible isoform of the NO synthase, can be triggered through several stimuli, including dsRNA, TNF-α, and proinflammatory cytokines [95,96,97]. iNOS expression is prominent in astrocytes and microglia, where it is induced by danger-associated signals or damage of the brain parenchyma. These extracellular pathogenic stimuli activate NF-κB, which in turn activates the NO/cGMP/cGK pathway by inducing iNOS expression. Through the upregulated iNOS expression, a large amount of NO molecules is produced, activating the astrocytic NO-GC and leading to the production of cGMP. An increase in cGMP upon iNOS expression has been shown in cell culture and acute brain slices [45,46,47,48,49,50,51,52,53,54,55,56,57,58,59,60,61,62,63,64,65,66,67,68,69,70,71,72,73,74,75,76,77,78,79,80,81,82,83,84,85,86,87,88,89,90,91,92,93,94,95,96,97,98]. The produced cGMP then activates cGK, thus activating the respective effector pathways.

Astrogliosis is a part of the immune response associated with a change in astrocytic morphology towards a hypertrophic state, a translocation towards the site of the immune challenge, and a change in gene expression. One important hallmark of astrocyte reactivity is an increase in the expression of the glial fibrillary acidic protein (GFAP). It was shown that an increase in cGMP leads to an increase in GFAP expression and that the induction of GFAP expression is dependent on NO/cGMP/cGK signaling [99,100,101,102]. Furthermore, cGMP can induce a change in astrocytic morphology via GFAP and actin remodeling. This process requires RhoA-GTPase to be phosphorylated, and thus inhibited, by cGK [101]. Although this is the most commonly described role of cGMP in astrocytes, there are also studies showing a decrease in GFAP expression upon administration of NO-GC stimulators [103]. Next to the signaling via NO, this pathway can also be triggered by ANP, since astrocytes contain pGC-A [45].

### 3.3. cGMP Signaling in Microglia

As the resident immune cells, microglia are the main mediators of inflammatory response in the brain. As in astrocytes, cGMP signaling is involved in microglial response to tissue damage or challenge with pathogens. Nevertheless, astrocytes and microglia show some differences in the cGMP signaling. First of all, microglia do not contain the Ca^2+^-dependent NOS, but the expression of iNOS can be induced by multiple cytokines, LPS, amyloids, or viral challenges [25,104,105,106]. The NO-activated NO-GC has not been detected in microglia; thus, in these cells, NO seems to serve a different function [25,107,108], for example, by attenuating the generation of reactive oxygen species and respiratory burst in a cGMP-independent manner [109]. In peripheral macrophages, activation of peroxisome proliferator-activated receptor γ by NO attenuates respiratory bursts [110]; whether a similar effect is present in microglia is not known.

As a diffusible molecule, NO has a second/retrograde messenger function in both intra- and intercellular signaling. For example, the NO produced in microglia during inflammation induces the expression of Nrf2 as well as heme-oxygenase 1 (see above), and thereby activates a feedback loop inhibiting the iNOS expression in an autocrine as well as paracrine (e.g., in the neighboring astroglia) way. Some studies revealed a correlation between an increase in microglial iNOS, neuronal cGMP, and neuronal apoptosis [111,112], while others showed that increased NO concentration is neuroprotective [113]. One theory explaining these different results suggests that the duration of exposure to the increased NO/cGMP/cGK signaling is relevant for neurotoxicity. Accordingly, transient exposure would have antiapoptotic and sustained exposure proapoptotic effects [111].

As microglia do not contain NO-GC (see above), they likely produce cGMP in a NO-independent but ANP-dependent manner [108]. Indeed, stimulation of microglial cells with ANP reduces LPS-induced microglial activation [114]. Nevertheless, several publications revealed microglial sensitivity to the manipulations of the NO-GC signaling [103,115]. For example, in the scratch wound in vitro models, as well as under acute spinal cord injury conditions in vivo, the application of the NO-GC inhibitor ODQ significantly reduced microglial motility [116,117,118,119].

Whether the cGMP signaling has a pro- or anti-inflammatory effect in microglia is still debated. In the case of traumatic brain injury, the application of a PDE inhibitor leads to a reduction in the number of amoeboid microglia around the site of injury, together with a decrease in the expression of the ionized Ca^2+^-binding adaptor molecule 1 (Iba-1) and Cd11b [102]. It was shown that cGMP can reduce the inflammatory response to LPS stimulation, as shown by the reduced expression of proinflammatory genes [103,114]. On the other hand, there are also reports of increased Iba-1, iNOS, Cd11, MHC class II, and proinflammatory cytokine expression (i.e., increased inflammation) upon cGMP accumulation [99,120,121]. The inflammation-induced deramification of microglia does not seem to be affected by cGMP signaling [122].

Although many studies on the role of cGMP in microglia have been conducted, the results remain controversial, suggesting a complex and context-dependent mechanism, which needs further investigation.

### 3.4. cGMP Signaling in Endothelial Cells and Pericytes

Endothelial cells are part of the blood–brain barrier (BBB). The NO/cGMP/cGK pathway in these cells is involved in cell motility, migration, proliferation, processes important for vessel integrity, endothelial barrier function, and angiogenesis [123]. All relevant proteins of the NO/cGMP/cGK pathway can be found in endothelial cells (Figure 2) [124]. For example, it was shown that CNP-pGC/cGMP signaling causes vasodilation in cerebral arterioles [125] and modulates the BBB permeability via a reduced expression of zonula occludens-1 protein forming tight junctions [126]. In the case of pathogen challenge, however, the signaling seems to work differently. Activated endothelial cells express the intercellular adhesion molecule (ICAM) and the vascular cell adhesion molecule (VCAM), which allow leukocytes to enter the brain parenchyma [127]. The cGMP pathway downregulates the expression of both proteins, leading to decreased immigration of peripheral immune cells (T-cells) into the brain [99]. In addition, the activation of the NO/cGMP/cGK pathway counteracts the inflammatory cytokine-induced increase in the BBB permeability, likely by affecting F-actin remodeling [128,129].

Pericytes are contractile cells on capillaries, which control cerebral blood flow and contribute to the BBB [130]. NO, acting on pericytes in a cGMP-dependent manner, can induce vessel dilation [131] since these cells contain both sGC [132] and cGK (Figure 2) [133]. The NO/cGMP/cGK-mediated dilation of blood vessels is caused by the relaxation of vascular smooth muscle cells either directly or via endothelial cells and pericytes [134,135]. One of the factors interconnecting endothelial cells and pericytes is the vasoconstrictive endothelin-1 (ET1) [136]. There is a strong interconnection between NO and ET1 signaling [137]. NO can inhibit ET1 release and ET1 can inhibit NO production [137,138]. Together, this allows for tight regulation of vasodilation.

## 4. Interplay between Calcium and cGMP Signaling

Calcium is the most abundant second messenger of the body and brain and an important mediator in a variety of cellular processes. Under physiological conditions, the [Ca^2+^]_i_ is low due to the constant removal of Ca^2+^ from the cytosol. The extrusion mechanisms include the plasma membrane Ca^2+^-ATPase (PMCA pump), the transmembrane Na^+^/Ca^2+^ exchanger, as well as the sarco-/endoplasmatic reticulum Ca^2+^-ATPase (SERCA pump), mediating the reuptake of Ca^2+^ into the endoplasmatic reticulum (ER). The Ca^2+^ and the cGMP signaling pathways are interconnected. The most obvious link is the Ca^2+^-dependent NO production by nNOS and eNOS described above. Both constitutively active forms of NOS have a binding site for calmodulin. For nNOS, it was shown that at low [Ca^2+^]_i_, calmodulin (CaM) is either free or pre-bound to nNOS with its C-terminus. Upon an increase in [Ca^2+^]_i_, also the N-terminus of CaM is binding to nNOS, enabling the activation of this enzyme. Upon sequestration of intracellular Ca^2+^, Ca^2+^ dissociation from the N-terminus of CaM likely causes nNOS deactivation [6]. Besides the direct Ca^2+^/CaM sensitivity of the nNOS activation process, there is also an additional Ca^2+^-dependent regulatory mechanism, mediated by the Ca^2+^/calmodulin-dependent kinase II (CaMKII) [139]. Phosphorylation of nNOS by the CaMKII attenuates the catalytic activity of nNOS and leads to a reduction in NO production [139,140]. Additionally, when Ca^2+^ and cGMP pathways are activated simultaneously, they can have a synergistic effect on the induction of c-Fos expression [141,142].

It must be taken into account that cGMP signaling is strongly compartmentalized. The site of production, either in the cytosol by NO-GC or at the membrane by pGC, is one of the factors generating a localized cGMP pool. Secondly, the spread of cGMP throughout the cell is limited by PDEs. PDEs themselves show distinct subcellular localization and thus, a differential control of the specific cGMP pools is possible (reviewed in [143]). Similarly, the interplay between the Ca^2+^ and cGMP signaling differs depending on the exact subcellular location. It was shown, for example, that cGMP produced in the cytosol enhances the SERCA-mediated reuptake of Ca^2+^ into the ER, thus reducing the [Ca^2+^]_i_, but downregulates the efflux of Ca^2+^ via PMCA [144]. The pGC signaling via CNP most likely stimulates L-type Ca^2+^ channels, thus increasing the influx of Ca^2+^ via the plasma membrane [145]. Additionally, cGMP was shown to control the release of Ca^2+^ from the ER, mediated by ryanodine receptors (RyR). In hippocampal LTP, for example, NO-mediated activation of the NO/cGMP/cGK pathway likely also activates ADP-ribosyl cyclase, stimulating the production of cyclic ADP-ribose, which acts synergistically with cytosolic Ca^2+^ to trigger RyR-mediated Ca^2+^ release from the intracellular Ca^2+^ stores [146].

Ca^2+^ signaling is an important mediator of neuro-glial interaction. The release of glutamate from neurons can trigger intercellular Ca^2+^ waves in astrocytes, thus altering synaptic transmission between neurons [147]. It was shown that in astrocytes, RyR-mediated Ca^2+^ release from the ER is potentiated by the activation of the NO/cGMP/cGK signaling pathway [148]. Such interplay between Ca^2+^ and NO supports the generation and propagation of both intra- and intercellular Ca^2+^ waves, as rises in [Ca^2+^]_i_ trigger the generation of NO by constitutively active NOSs, and NO, as a freely diffusible molecule, supports RyR-mediated Ca^2+^ release in further cells/subcellular compartments. Finally, rises in [Ca^2+^]_i_ were shown to release ANP from astrocytes by means of Ca^2+^-dependent exocytosis [149]. ANP, in turn, might contribute to several key regulatory processes within the neuro-glial network, including the activity-dependent regulation of cerebral blood flow or activation of microglia [150,151] (Figure 2).

Taken together, the interplay between the Ca^2+^ and cGMP signaling forms a closed loop, important for the proper function of neuro-glial networks. Graded rises in [Ca^2+^]_i_ result in the graded NO/cGMP production and, via cGMP-mediated Ca^2+^ influx and efflux from the cytosol, control the duration and the activation strength of downstream effector pathways. A disruption of this fine regulatory mechanism has a strong functional impact on endothelial and glial cells as well as neuronal signaling (Figure 2), making a further understanding of this interconnection highly relevant.

## 5. cGMP in Alzheimer’s Disease

Alzheimer’s disease is a neurodegenerative disorder characterized by a progressive loss of memory and a reduced capability for learning and memory formation. The main histopathological hallmarks of AD include the formation of amyloid plaques, accumulation of phosphorylated protein tau, formation of neurofibrillary tangles, neuroinflammation, and profound neuronal loss. Although AD pathology was first described in the early 20th century, the precise disease mechanism remains elusive. The amyloid hypothesis of AD posits that overproduction of amyloid-β (Aβ), produced by sequential cleavage of amyloid precursor protein (APP) by the β- and γ-secretases, leads to the formation of toxic Aβ species (oligomers and fibrils) precipitating in Aβ-plaques. According to this hypothesis, the increased production of self-aggregating forms of Aβ (e.g., Aβ42), due to missense mutations in the APP or other AD-related genes or failure in Aβ clearance, causes the gradual deposition of Aβ oligomers as diffuse plaques, microglial and astrocytic activation, oxidative injury, and formation of neurofibrillary tangles built of the hyperphosphorylated tau protein, together leading to widespread synaptic dysfunction and neuronal loss [152,153].

Having detrimental effects at high concentrations, at low concentrations, Aβ is important for memory formation (reviewed in [154]) and is known to be released in vivo in an activity-dependent manner [155]. It was shown that cGMP enhances the colocalization of APP with beta-site APP cleaving enzyme 1 (BASE-1) in endolysosomes, thus increasing the production of Aβ at synapses [156,157]. Furthermore, Aβ is needed for the cGMP-induced effects on LTP [157], known to be impaired in AD [158,159]. Next to synaptic dysfunction, Aβ can sustain low-grade neuroinflammation [160,161]. This inflammation can become neurotoxic, causing a disruption of neuronal signaling and eventually neuronal death [162,163]. Since cGMP signaling is essential for LTP and cGMP is a part of the inflammatory response (see above), deregulation of cGMP signaling likely contributes to the development of AD. Indeed, several studies succeeded in identifying AD-related changes in the NO/cGMP/cGK pathway (Figure 3). The first change regarding the cGMP signaling, identified in AD, was an increased expression of eNOS in neurons as well as nNOS in neurons and reactive astrocytes [164]. The upregulation of nNOS is associated with neuronal loss in the AD brain (Figure 3) [164,165]. It was also shown that the prolonged overactivation of nNOS induces synapse loss in motor neurons via the NO/cGMP/cGK pathway [93]. Differently from nNOS, eNOS in the AD brain seem to be downregulated in neurons (Figure 3) [166] but upregulated in astrocytes [18]. Interestingly, changed nNOS expression was shown to counteract dysregulated synaptic Ca^2+^ signaling at the early stages of amyloid deposition in animal models of AD [16]. In general, the vicinity of amyloid plaques is characterized by Ca^2+^ store-mediated neuronal hyperactivity [8,9,167]. By reducing presynaptic RyR-mediated Ca^2+^ release, NO signaling reduced synaptic transmission, thus counteracting neuronal hyperactivity [16].

As iNOS expression is associated with inflammation, and neuroinflammation is a major hallmark of AD, its expression is high in AD brains [168]. Contrary to the control aged brains, in AD brains, iNOS expression is induced not only in glial cells but also in neurons (Figure 3) [169,170]. Since iNOS is constitutively active, it produces large amounts of NO. A possibly toxic mechanism caused by NO overproduction is the accumulation of peroxynitrite (ONOO^−^), through a reaction of NO with superoxide [171]. ONOO^−^ can cause nitrotyrosylation of proteins, leading to their dysfunction [25,172]. This effect takes place in animal models of AD as well as in AD brains [173,174]. Besides being a reactive nitrogen species itself, peroxynitrite can generate reactive oxygen species (ROS) [175], thus promoting mitochondrial dysfunction and cytotoxicity [25,172]. The reduction in ROS, for example, through synthetic or plant-based antioxidants, was shown to have anti-inflammatory and neuroprotective effects [176]. Therefore, future AD treatment might benefit from a combination of antioxidant treatment with modifiers (see below) of the cGMP signaling pathway.

The activity of NO-GC decreases already in the aging brain [177] and is further downregulated in AD (Figure 3) [178]. Some inflammatory factors, such as IL-1β, LPS, and Aβ, can downregulate the expression, activity, and availability of NO-GC [179,180]. This effect was most prominent in human astrocytes located close to amyloid plaques [181]. Moreover, Aβ seems to inhibit NO-GC activity in microglia via binding to the cell surface receptor cluster of differentiation 36 (CD36) [115]. In neurons, so far, no AD-related downregulation of NO-GC was observed [181].

Little is known about the AD-related changes in NP-mediated signaling. However, in AD patients, the BNP levels were reported to decrease [38,182]. Additionally, Mahinrad et al. reported a significant increase in the NPR-A protein level in the brains of AD patients [38]. For ANP and CNP as well as NPR-C, there was a trend toward lower expression, which, however, did not reach the level of statistical significance [38]. PDEs are the third factor influencing the level of cGMP. No change in the expression levels of PDE2, 5, and 9 mRNA was found (Figure 3) [62], suggesting that the degradation of cGMP is not affected in AD. As for cGK, it was recently shown that it phosphorylates tau at Ser214 but not at Ser202, thereby likely reducing its pathological aggregation and shifting the protein from a pro-aggregant to a neuroprotective anti-aggregant conformation [183].

Although, as described above, many studies have been conducted on the proteins belonging to the NO/cGMP/cGK pathway, little is known about the overall AD-related changes in the cGMP concentration. In rat AD models, a reduction in the hippocampal cGMP concentration was identified [184]. Similarly, the severity of disease symptoms correlated with the severity of the cGMP reduction in the CSF, whereas the CSF cAMP levels did not change significantly [185,186]. In a mouse model of AD, the ability of the intrahippocampal NMDA infusion to stimulate cGMP production (measured by microdialysis) was significantly reduced already in young, amyloid-predepositing mice [187]. In contrast to WT animals, in AD mice, cGMP levels did not return to baseline but stayed elevated over a prolonged period of time. This suggests an impairment of the cGMP degradation by PDEs. So far, however, there are no data on AD-related changes in the intracellular cGMP concentration, but since the cumulative evidence points towards a decrease in the brain level of cGMP, PDE5 inhibitors have been tested as an AD treatment option. A potent PDE5 inhibitor, Sildenafil, able to cross the BBB, improved cognitive and memory functions in healthy rodents ([188,189,190] but see [191]). In Tg2576 mice, prolonged application of Sildenafil rescued cognitive impairment and lowered tau phosphorylation, with no change in Aβ load [192]. In APP/PS1 mice, the application of Sildenafil also resulted in improved cognitive performance. In contrast to Tg2576 animals, there was a reduction in soluble Aβ, which correlated with an increase in the phosphorylated CREB [191,193]. Furthermore, Sildenafil treatment reduced astrogliosis and microglia activation in several models of inflammation [194,195,196]. During inflammation, astrocytes increase the magnitude of ATP-induced Ca^2+^ signals, which was normalized by Sildenafil treatment [197].

In healthy young humans, no beneficial effect of Sildenafil on short-term memory was observed, besides a decrease in the reaction time [198]. Similarly, Sildenafil did not induce any significant improvement in auditory selective attention or verbal recognition memory [199]. However, in a recent drug repurposing study using an in silico approach based on insurance data from 7.23 million individuals, Sildenafil was identified as a target drug for AD treatment. In a 6-year-long follow-up, it was shown to reduce the risk of developing AD by 69% [200]. Moreover, recent pilot studies (10–20 patients) using a single dose (50 mg) of Sildenafil reported a Sildenafil-mediated decrease in the neural hyperactivity in the hippocampus and an increased cerebral metabolic rate of oxygen as well as cerebral blood flow [201,202]. Another PDE5 inhibitor, Tadalafil, has a longer half-life and better safety in chronic treatment [203], but lower capability to cross the BBB. However, during chronic administration of Tadalafil in a J20 AD model, sufficient PDE inhibition was found within the brain [204]. In this study, Tadalafil rescued cognitive deficits and decreased tau hyperphosphorylation but did not affect Aβ load [204]. Furthermore, in acute slice experiments, Tadalafil was able to improve LTP in APP/PS1 mice [204].

Interestingly, PDE5 inhibitors can also induce autophagy [194,205]. Among others, autophagy is regulated via 8-nitro-cGMP (a product of the chemical reaction between NO and cGMP) and mTOR (mammalian target of rapamycin) and is essential for the clearance of aggregated Aβ and hyperphosphorylated tau [206,207]. PDE5 inhibitors induced autophagy by downregulating mTOR or by activating 5′ AMP-activated protein kinase [194,205]. Next to synthetic PDE inhibitors, there are also some natural compounds able to modulate cGMP signaling [208]. The flavonoid Icariside, obtained from epimedium brevicornum, and its derivate Icariin can inhibit PDE5 (Figure 3). Treatment of mouse (APP/PS1 animals) and rat (Aβ injection in the hippocampus) models of AD with these compounds decreased expression of APP, Aβ, and PDE5 and significantly attenuated AD-related cognitive deficits [209,210]. Further natural compounds, able to inhibit PDEs (for more details see [208]), were so far not tested in the preclinical AD models.

Recently, brain penetrant NO-GC stimulators have been developed as targets for AD treatment [103,211]. In vitro and in vivo studies with CYR119 revealed anti-inflammatory effects in murine microglia cell culture and the intact brain [103]. Furthermore, the BBB penetrant NO-GC stimulator CY6463 improved LTP in a Huntington’s disease mouse model and reduced loss of neuronal spines in a mouse model of AD (APP/PS1 mice). CY6463 was also able to improve cognitive function and reduce inflammatory markers in the brain [211]. So far, no results on clinical studies for the usage of these compounds in AD have been released (the first data of the phase 2 study on CY6463 are expected in July 2022; ClinicalTrials.gov Identifier: NCT04798989), but the preclinical studies show a great potential for sGC activators in AD treatment (Figure 3).

Taken together, the preclinical studies conducted so far point towards deregulation of the NO/cGMP/cGK signaling pathway in AD. However, as illustrated in Figure 3, this deregulation is complex and cell type-specific, and the extent as well as the direction of the change need further investigation.

## 6. Concluding Remarks and Future Directions

cGMP signaling controls the function of major cell types of the brain parenchyma and the vasculature both alone and in an interplay with intracellular Ca^2+^ signaling. Indeed, Ca^2+^ enhances NO production and increases the activity of the cGMP pathway. On the other hand, cGMP influences [Ca^2+^]_i_ by modulating the release of Ca^2+^ from the intracellular stores and the Ca^2+^ influx/efflux over the cell membrane. Although it is known that both signaling pathways are deregulated in AD, the interplay between the two has been largely neglected.

Given the complexity of the molecular mechanisms described above as well as the cellular players involved, this interplay must be studied in vivo. Current developments in the field of optical Ca^2+^ and cGMP sensors [212,213,214] enable longitudinal in vivo imaging studies, allowing for intracellular data on [Ca^2+^]_i_ and cGMP at the subcellular resolution and over different stages of AD. This is extremely important because early neuronal hyperactivity [8,215] likely increases not only [Ca^2+^]_i_ but also NO and cGMP production, and decreased cGMP could enhance neuronal hypoactivity, found in late-stage AD [216,217].

Depending on the exact outcome of the longitudinal preclinical studies, individualized treatment options taking into account the sex (AD has a higher prevalence in women compared to men), disease stage, and the level of neuroinflammation can be developed. Fortunately, future therapies can choose from a rich cGMP-modulating toolkit containing BBB-permeant PDE inhibitors (e.g., Sildenafil, Tadalafil), NO-GC stimulators (e.g., CYR119, CY6463), etc. The hypothesis-driven development of such therapeutics is, however, impossible without a clear understanding of the direction and the extent of the modulation of NO/cGMP/cGK signaling pathway in aging and AD and its cell type-specific interplay with the intracellular Ca^2+^ signaling.

## Figures and Tables

**Figure 1 ijms-23-07048-f001:**
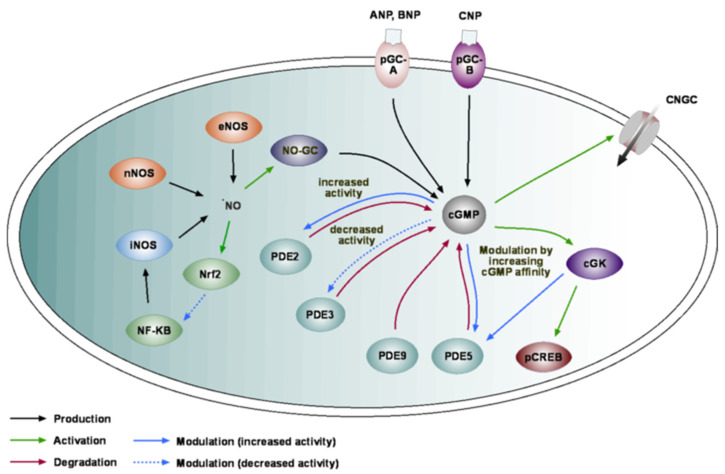
cGMP signaling pathway. cGMP production is induced via activation of either NOS or natriuretic peptides (NPs). NO activates NO-GC in the cytosol, and pGC-A and pGC-B produce cGMP in the membranous fraction. cGMP stimulates cGK, which phosphorylates target proteins. cGMP is degraded by PDEs. PDE5 and 9 are cGMP-specific, and PDE1, 2, 3, and 10 are dual-substrate PDEs, degrading both cAMP and cGMP. NOS: nitric oxide synthase; iNOS: inducible NOS; eNOS: endothelial NOS; nNOS: neuronal NOS; NO-GC: soluble guanylate cyclase; Nrf2: Nuclear factor erythroid 2-related factor 2 cGMP: cyclic guanosine-3′,5′-monophosphate; NF-κB: nuclear factor κB; ANP: atrial natriuretic peptide; BNP: brain natriuretic peptide; CNP: C-type natriuretic peptide; pGC-A: particulate guanylate cyclase A; pGC-B: particulate guanylate cyclase B; cGK: cyclic guanosine monophosphate-dependent protein kinase, also known as protein kinase G (PKG); PDE: phosphodiesterase; CNCG: cyclic nucleotide-gated channel; pCREB: phosphorylated cyclic adenosin monophosphate (cAMP) response element-binding protein.

**Figure 2 ijms-23-07048-f002:**
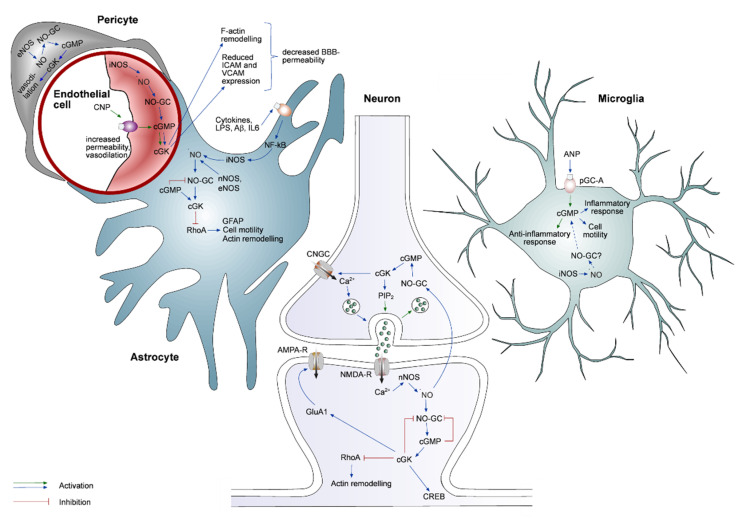
The functional roles of the cGMP signaling pathways in neurons, astrocytes, microglia, endothelial cells, and pericytes (see text for detailed description). AMPA: α-amino-3-hydroxy-5-methyl-4-isoxazolepropionic acid receptor; GluA1: ionotropic glutamate receptor of AMPA type, subunit 1; CREB: cAMP response element-binding protein; GFAP: glial fibrillary acidic protein; IL6: interleukin 6; LPS: lipopolysaccharide; NF-κB: nuclear factor kappa B; NMDA-R: *N*-methyl-*D*-aspartate-receptor; RhoA: Ras homolog family member A; ICAM: intercellular adhesion molecule; VCAM: vascular cell adhesion molecule; all other abbreviations are explained in the figure legend for Figure 1.

**Figure 3 ijms-23-07048-f003:**
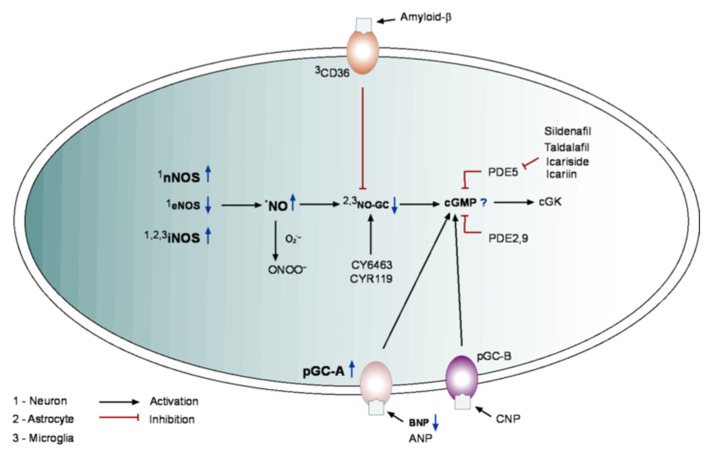
Changes in cGMP-mediated signaling in Alzheimer’s disease. The levels of nNOS, iNOS, pGC-A, and NO are increased, and the activity of NO-GC is reduced. Furthermore, amyloid-β inhibits NO-GC activity. Whereas the cGMP concentration in the CSF is reduced, the mean effect on the intracellular cGMP level is unknown. Possible targets for AD treatment are the stimulation of NO-GC and the inhibition of PDE inhibitors (see text for further explanations). CD36: cluster of differentiation 36; ONOO^−^: peroxynitrate, all other abbreviations are explained in the figure legend for Figure 1.

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
