# Peer review of "The Interplay between cGMP and Calcium Signaling in Alzheimer’s Disease"

_ijms, 2022, doi:10.3390/ijms23137048_

Round 1
Reviewer 1 Report
I have some major comments on this manuscript.
- Write the limitations of your manuscript at the end of the abstract section.
- Discuss some key features of your manuscript in the abstract section.
- Include some more signaling components in figure 1. Relate NF-kB with NRf2 in this figure.
- How autophagic factors are related to cGMP and calcium signaling in AD pathogenesis and treatment.
- Figure 3 is very superficial, include some additional components in this figure.
- A table regarding the role of the medicinal plant of cGMP and calcium signaling in AD will improve the quality of your manuscript. Likewise, modify the title of your paper accordingly.
- Complete editorial checking will be needed to correct the grammatical and punctuation mistakes.
- The discussion and conclusion section is missing in the manuscript.
- Cite PMID: 30605887, PMID: 30503937 in the Alzheimer’s disease portion of your manuscript. Relate the role of several transcription factors with cGMP and calcium signaling.
- Check every reference in your manuscript, some texts are not related to their sentence.
Author Response
Reply to the reviewers’ comments
We would like to thank the reviewers for their comments and for their willingness to improve the manuscript. Answers to the specific points of reviewers are given below.
Referee #1:
I have some major comments on this manuscript.
- Write the limitations of your manuscript at the end of the abstract section.
We are sorry but we find it difficult to talk about the limitations in the context of a review, not a research article. We tried to reflect the current state of the knowledge and to summarize the current literature to the best of our knowledge. In response to the comment of the reviewer, we did modify the abstract to stress the limitations of the current knowledge about the interplay between the cGMP and calcium signaling.
- Discuss some key features of your manuscript in the abstract section.
Thank you. Done.
- Include some more signaling components in figure 1. Relate NF-kB with NRf2 in this figure.
We thank the reviewer for this suggestion. In response to this comment, we have added the interaction between iNOS, NF-κB and Nrf2, providing a feedback loop for NO-coupled reduction in iNOS expression during inflammation (see Figure 1; p. 2,3 and 7).
- How autophagic factors are related to cGMP and calcium signaling in AD pathogenesis and treatment
Thank you for this suggestion. We added the literature on how autophagy is impaired and how PDE5 inhibitors can induce autophagy (see p. 12).
- Figure 3 is very superficial, include some additional components in this figure.
Thank you for this suggestion. We intended to keep the figure simple so that major AD-related changes in the cGMP signaling pathway are visible at first glance. Following the reviewer’s comment, we have added the pathway-specific treatment options, which are currently under development, including the therapeutics derived from medicinal plants.
- A table regarding the role of the medicinal plant of cGMP and calcium signaling in AD will improve the quality of your manuscript. Likewise, modify the title of your paper accordingly.
We included references discussing the use of the PDE5 inhibitors derived from medicinal plants (see p. 12). Since the medical plants are not the main focus of this review, and, to our knowledge, there are no further studies describing the role of medicinal plants in the regulation of an interplay between calcium and cGMP, we refrained from including the medical plant table into the revised version of the manuscript.
- Complete editorial checking will be needed to correct the grammatical and punctuation mistakes.
Thank you, corrected.
- The discussion and conclusion section is missing in the manuscript.
Thank you for this suggestion. We have added a section “concluding remarks and future directions” at the end of the manuscript (see p.12). However, as this is a review article, we are somewhat puzzled by the request to add the discussion section… In our eyes, all sections other than the introduction section belong to the discussion.
- Cite PMID: 30605887, PMID: 30503937 in the Alzheimer’s disease portion of your manuscript. Relate the role of several transcription factors with cGMP and calcium signaling
We thank the reviewer for suggesting these interesting papers. We added information on the possible use of antioxidants in AD treatment to the revised version of the manuscript (see p. 10).
- Check every reference in your manuscript, some texts are not related to their sentence.
Thank you, corrected.

Reviewer 2 Report
This is a wonderful review of the literature and a very important addition to the literature. My issues are with the last paragraph.
The authors wrote “Taken together, conducted so far preclinical studies clearly pointed towards deregulation of the NO/cGMP/cGK signaling pathway in AD. In clinical trials, however, the use of PDE inhibitors aiming at increasing the cGMP levels so far did not yield the expected improvement of learning and memory function.”
Briefly, please provide your true assessments about why these preclinical studies have not been successful. There must be additional reasons such as excessive breakdown of cGMP, inadequate access of test drugs to the cortex, etc etc that may explain the situation. Do you have any ideas as to how treatment strategies could be improved?
The authors wrote, “Still, recent results on long-term usage of Sildenafil for different purposes reveal its potential for preventing AD [199],” (please replace comma with period).
Please expand on this a little, what were the recent results and their potential value??
The authors wrote, “ The newly developed NO-GC activators further broaden the therapeutic toolkit and hold potential as future AD therapeutics. The hypothesis-driven development of such therapeutics is, however, impossible without a clear understanding of the details of NO/cGMP/cGK signaling pathway in aging and AD.
Please provide brief text about what experimental details of NO/cGMP/cGK signaling pathway in aging and AD would be valuable.
Author Response
Reply to the reviewers’ comments
We would like to thank the reviewers for their comments and for their willingness to improve the manuscript. Answers to the specific points of reviewers are given below.
Referee #2:
This is a wonderful review of the literature and a very important addition to the literature. My issues are with the last paragraph.
We thank the reviewer for the detailed and benevolent assessment of our work and for the willingness to help us to improve the manuscript.
The authors wrote “Taken together, conducted so far preclinical studies clearly pointed towards deregulation of the NO/cGMP/cGK signaling pathway in AD. In clinical trials, however, the use of PDE inhibitors aiming at increasing the cGMP levels so far did not yield the expected improvement of learning and memory function.”
Briefly, please provide your true assessments about why these preclinical studies have not been successful. There must be additional reasons such as excessive breakdown of cGMP, inadequate access of test drugs to the cortex, etc etc that may explain the situation. Do you have any ideas as to how treatment strategies could be improved?
We thank the reviewer for this comment. Because the most recent human studies, which we have now explained in more detail in the revised version of the manuscript (see p. 11 nad 12), are more positive, in the revised version of the manuscript we decided to put less emphasis on the failures of older clinical trials. Improved treatment strategies, to our opinion, must take into account the patient’s sex, disease stage, the level of neuroinflammation as well as the new preclinical data about the cell-type-specific dysregulation of Ca2+- and cGMP-signaling. We have included this discussion in a new “Concluding remarks and future directions” section (p. 12).
The authors wrote, “Still, recent results on long-term usage of Sildenafil for different purposes reveal its potential for preventing AD [199],” (please replace comma with period).
Please expand on this a little, what were the recent results and their potential value??
Done, thank you (see p. 11).
The authors wrote, “ The newly developed NO-GC activators further broaden the therapeutic toolkit and hold potential as future AD therapeutics. The hypothesis-driven development of such therapeutics is, however, impossible without a clear understanding of the details of NO/cGMP/cGK signaling pathway in aging and AD.
Please provide brief text about what experimental details of NO/cGMP/cGK signaling pathway in aging and AD would be valuable.
We thank the reviewer for this suggestion. In the “concluding remarks and future directions” section (p.12) , we added a discussion on the knowledge required for broadening our understanding of cGMP pathology in AD.

Round 2
Reviewer 1 Report
Manuscript is not revised properly. Fundamental flaws still found in methods and result section. Discuss are not well correlated. Manuscript is not suitable for publication.
Author Response
We are sorry, we do not understand the statement of the reviewer, as our review article contains neither methods nor results sections.